# Goal Conditioned Reinforcement Learning for Photo Finishing Tuning

**Jiarui Wu**[1,2*], **Yujin Wang**[1†], **Lingen Li**[1,2], **Fan Zhang**[1], **Tianfan Xue**[2]

[1]Shanghai AI Laboratory  [2]The Chinese University of Hong Kong

`{wj024,tfxue}@ie.cuhk.edu.hk, lgli@link.cuhk.edu.hk,`
`{wangyujin,zhangfan}@pjlab.org.cn`

## Abstract

Photo finishing tuning aims to automate the manual tuning process of the photo finishing pipeline, like Adobe Lightroom or Darktable. Previous works either use zeroth-order optimization, which is slow when the set of parameters increases, or rely on a differentiable proxy of the target finishing pipeline, which is hard to train. To overcome these challenges, we propose a novel goal-conditioned reinforcement learning framework for efficiently tuning parameters using a goal image as a condition. Unlike previous approaches, our tuning framework does not rely on any proxy and treats the photo finishing pipeline as a black box. Utilizing a trained reinforcement learning policy, it can efficiently find the desired set of parameters within just 10 queries, while optimization-based approaches normally take 200 queries. Furthermore, our architecture utilizes a goal image to guide the iterative tuning of pipeline parameters, allowing for flexible conditioning on pixel-aligned target images, style images, or any other visually representable goals. We conduct detailed experiments on photo finishing tuning and photo stylization tuning tasks, demonstrating the advantages of our method. Project website: https://openimaginglab.github.io/RLPixTuner/.

## 1 Introduction

Image processing pipelines (ISPs) are widely used by photographers and artists to retouch images to match their desired appearance. Existing pipelines like Adobe Lightroom and Darktable allow users to interactively tweak meaningful sliders such as exposure, white balance, and contrast, which control the pipeline to perform a series of non-destructive edits to the input image. Though users can manually tune the slider parameters, it is laborious and time-consuming even for experienced experts. To this end, automatic photo finishing algorithms have been introduced to automate the process and have drawn growing attention in the community [11].

In this work, we aim to design an automatic tuning algorithm for black-box non-differentiable image processing pipelines. Although some preliminary research tries to propose fully differentiable image processing pipelines [11, 7] or approximate the existing pipelines using neural networks [29], they only support a limited set of image processing operations. On the other side, most commercial image processing pipelines, like Adobe Lightroom, are still black-box and non-differentiable, with a set of tunable parameters exposed to users. Under this setup, the goal of pipeline tuning is to automatically find the set of optimal parameters to achieve a desired image appearance, named the tuning target. More specifically, in this work, we study two different tuning targets. One is a target image with the same content as the input, but with a different rendering style, and we call it *photo finishing tuning*. The other is a target style image with different content, and the algorithm is to render the target in a similar way as the style target, and we call that *photo stylization tuning*.

---

[*]This work was done while Jiarui Wu interned at Shanghai AI Laboratory.
[†]Corresponding authors.

38th Conference on Neural Information Processing Systems (NeurIPS 2024).

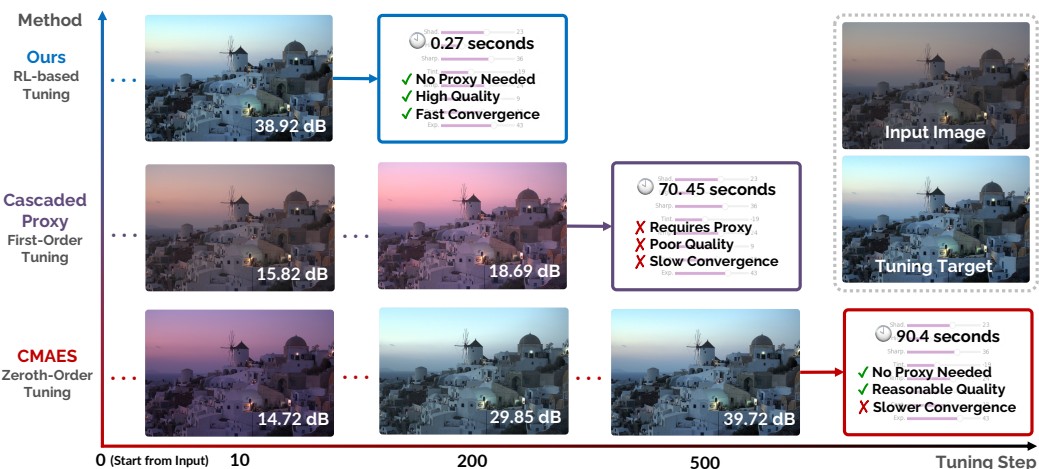

Figure 1: In this work, we propose an RL-based photo finishing tuning algorithm that efficiently tunes the parameters of a black-box image processing pipeline to match any tuning target. The RL-based solution (top row) takes only about 10 iterations to achieve a similar PSNR as the 500-iteration output of a zeroth-order algorithm (bottom row). Our method demonstrates fast convergence, high quality, and no need for a proxy.

One previous solution for photo finishing tuning is to use zeroth-order or first-order optimization. However, both of them are either time-consuming or limited to a small set of parameters. Zeroth-order optimizations [9, 21, 20, 4] are gradient-free searching methods and thus are normally very slow when the search space increases. First-order optimizations [28, 32, 29] accelerate the searching process using gradient descent, but they either require the processing pipeline itself to be differentiable, or a neural proxy is pre-trained to find a differentiable proxy of the original pipeline. For a complex commercial imaging pipeline, like cellphone camera pipelines, this proxy may not fully reproduce original pipelines [28]. Neural photo-finishing [29] improves the proxy accuracy by breaking a complex pipeline into small modules, but this does not apply to black-box or dynamically reconfigurable pipelines [32]. Considering all these limitations, this brings a challenging question: Is there an efficient parameter-searching algorithm that is applicable to non-differentiable finishing pipelines?

To solve this challenge, we propose a novel goal-conditioned reinforcement learning (RL) approach dedicated to photo finishing tuning. At each RL iteration, the policy network takes the tuning target and the currently tuned image as input, and finds a better set of parameters that makes the finishing results closer to the target. This RL-based searching algorithm has several advantages over traditional optimization methods. Compared with the zeroth-order solution, the RL policy can more accurately predict the potential searching direction, while zeroth-order searching can only rely on less effective tries. As a result, RL searching is much more efficient. As shown in Fig. 1, the RL-based solution (top row) only takes about 10 iterations to reach a similar PSNR as the 500-iteration output of a zeroth-order algorithm (bottom row). Also, compared with first-order optimization, RL-based tuning directly optimizes the non-differentiable image processing pipeline without a differential proxy. Therefore, it is not limited by the variety and complexity of image processing operations and pipelines, achieving much better tuning results. As shown in Fig. 1, RL-based tuning reaches 38.92dB (top row), while the first-order solution only obtains 18.69dB (middle row).

To train an efficient RL policy for tuning, we also propose a novel state representation dedicated to this task. The state representation should model the relationship between the photo editing space and our policy. To achieve that, our state representation consists of three key components: a CNN-based feature representation to encode global and local features, a photo statistics representation to match the photographic statistics between the input and the goal, and an embedding of historical actions. These representations better fit the RL policy into our task and guide the policy to generate the next set of parameters effectively. Lastly, we also design reward functions for both photo finishing tuning and photo stylization tuning, enabling our framework to tune photos to different targets.

We validate the effectiveness of our framework with extensive experiments on both photo finishing tuning and photo stylization tuning. Our RL-based framework significantly outperforms previous methods in both tasks in terms of both efficiency and image quality. Experimental results demonstrate

that our goal-conditioned policy is an efficient photo finishing tuner capable of performing fine-grained control on image processing pipeline parameters to achieve various goals.

## 2 Related Work

**ISP tuning.** Recent developments in end-to-end AI-based ISP pipelines show potential as alternatives to traditional mobile ISPs [13, 14, 25]. Yet, traditional parametric ISPs remain preferred in consumer cameras for their controllability, efficiency, and interpretability. These systems require expert-driven, labor-intensive tuning to enhance image quality and achieve desired aesthetic effects. Efforts to streamline this process are ongoing, aiming to reduce the need for manual adjustments. Gradient-based optimization is a notable strategy in this area. Tseng et al. [28] have applied differentiable black-box proxies to simplify ISP tuning. Further research by Tseng et al. [29] has opened up the ISP pipeline into white-box manageable modules, learning differentiable proxies for each module and subsequently integrating them, which improves the tuning performance. Additionally, Qin et al. [23] introduced an attention-based CNN approach for scene-aware ISP tuning. However, this method lacks integration of sequence-specific prior knowledge. They later developed a framework for predicting ISP hyper-parameters sequentially [24], optimizing parameters based on their relationships and similarities. In a different approach, Mosleh et al. [20] utilized a genetic evolutionary algorithm with a zero-order stochastic solver[9] to directly optimize hardware-specific image processing pipelines, circumventing the constraints of gradient-based methods. Moreover, Nishimura et al. [21] explored derivative-free optimization, employing nonlinear techniques and automatic reference generation for effective automation of image quality adjustments. Despite these advancements, the complexity of the image processing pipeline and the vast parameter space continue to challenge the efficiency of tuning methods.

**Photo stylization.** Automating image stylization, which is straightforward for humans, poses challenges for machines. Significant research has been conducted to bridge this gap. Karras et al. [15] pioneered StyleGAN, a network manipulating the latent space to control image styles at various scales. Building on this, Brooks et al. [2] developed InstructPix2Pix, which allows users to guide the stylization process through textual instructions, enhancing user-machine interaction. However, these methods often lack explainability and may alter the original image content. Further exploring transparency, Hu et al. [11] proposed a reinforcement learning-based white-box photo-finisher, though its need for explicit gradients limits compatibility with traditional systems. Kosugi et al. [18] addressed style diversity using unpaired data based on reinforcement learning. Despite these advances, these methods are restricted to a single style during training, limiting the user's ability to control the pipeline to produce images of any desired style during inference. Moreover, Tseng et al. [29] optimized proxy networks for image processing modules via a style loss function, achieving promising results but facing limitations in handling complex modules and style variability during inference.

## 3 Method

### 3.1 Problem Definition

Throughout this paper, we aim to tackle the photo finishing tuning problem. Given an input image $I_0$, an image processing pipeline $f_{\text{PIPE}}$ maps the input to a finished image $I_{\text{FINISHED}} = f_{\text{PIPE}}(I_0, P)$, controlled by a set of parameters $P$. Our task is to solve the inverse problem: given an input image $I_0$ and a tuning target $I_g$ (the goal condition in the RL framework), how to find the parameters that reach this target:

$$\arg \min_P \mathcal{L}(I_g, f_{\text{PIPE}}(I_0, P)). \tag{1}$$

The goal image $I_g$ can vary between different tasks. For photo finishing tuning, the goal image shares the same content as the input, and the tuning target is to minimize the distance between the pipeline output and the goal image $I_g$. For photo stylization tuning, the goal image is a style target with different content than the input, and the tuning target is to generate an output that matches the style of the goal image. Note that unlike artistic style transfer [6], we focus on photorealistic style transfer [31], where the processing pipeline does not change the content of an image.

### 3.2 Goal Conditioned Reinforcement Learning

We are the first to introduce a reinforcement learning (RL) approach to the photo finishing tuning task by formulating it as an end-to-end policy learning problem. Inspired by human experts who use

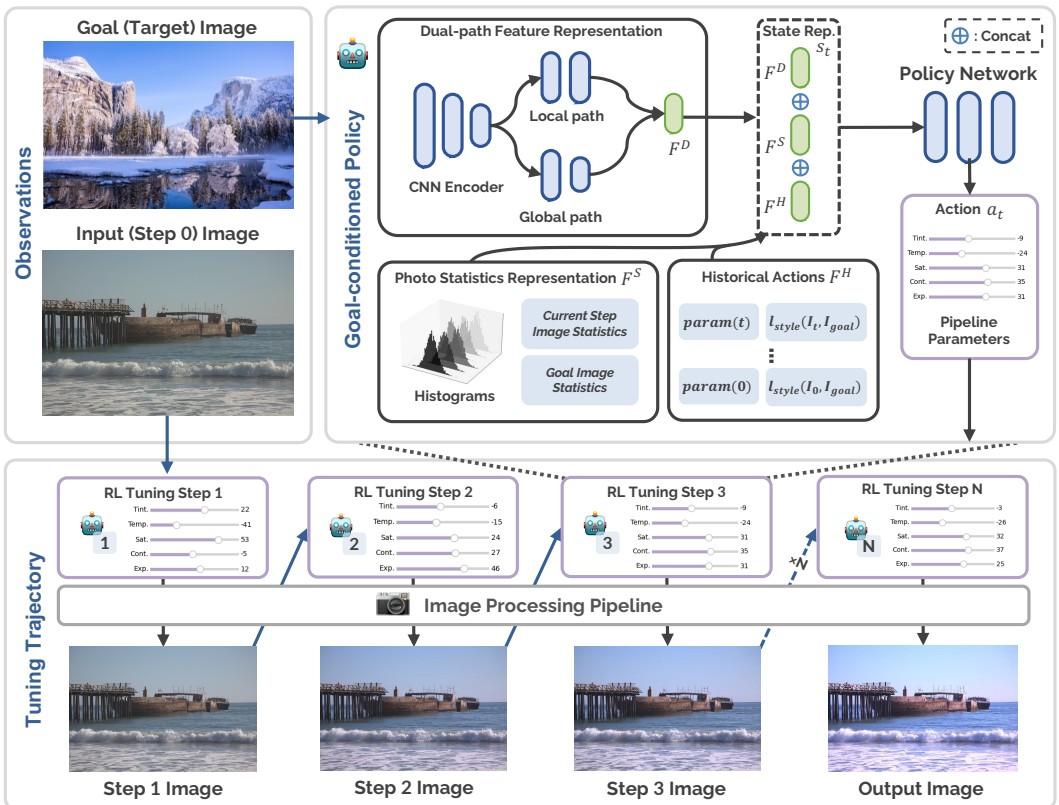

Figure 2: The overall framework. Top row: at each step, our policy maps the current image and the goal image to action (new parameters), with the help of our state representation consisting of dual-path features, photo statistics, and historical actions. Bottom row: visualization of iterative tuning trajectory of our RL-based photo finishing framework.

iterative trial and error in photo finishing, our method models the task as a decision process with multi-step feedback, akin to a Markov Decision Process. As shown in the bottom row of Fig. 2, our policy iteratively tunes the parameters of a given image processing pipeline to match a goal image. At each step, it takes the currently retouched image and the goal image as inputs, and then outputs the next action. This RL-based framework efficiently predicts pipeline parameters, requiring only minimal iterations (e.g., 10 queries to the image processing pipeline). Additionally, since RL optimization bypasses the need for gradient flow from the image processing pipeline, our system can handle any black-box pipeline, regardless of complexity.

The RL process is formally defined as follows. Let $\mathcal{S}$ be the state space, $\mathcal{O}$ be the observation space, $\mathcal{A}$ be the action space, $\mathcal{T}$ be the transition function, $\mathcal{R}$ be the reward function, $\mathcal{G}$ be the goal distribution, $\rho_0$ be the initial state distribution, and $\gamma$ the discount factor. All these forms a Goal-conditioned Partially-Observed Markov Decision Process $(\mathcal{S}, \mathcal{O}, \mathcal{A}, \mathcal{T}, \mathcal{R}, \mathcal{G}, \rho_0, \gamma)$ [19].

In each tuning episode $t$, the agent is given a goal image $I_g \in \mathcal{G}$, as well as an observation $o_t \in \mathcal{O}$ consists of the retouched image $I_t$ at step $t$ along with all historical actions and observations. The action $a_t$ is the parameter set $P$ used by image processing pipeline to generate the output image at the next step. And the transition function $\mathcal{T} : \mathcal{S} \times \mathcal{A} \rightarrow \mathcal{S}$ is the image processing pipeline $f_{\text{PIPE}}$ defined in Sec. 3.1. The reward function is $\mathcal{R}(s, I_g)$ for $s \in \mathcal{S}$ and $I_g \in \mathcal{G}$. We aim to learn a goal-conditioned policy $\pi(a|o, I_g) : \mathcal{S} \times \mathcal{G} \rightarrow \mathcal{A}$ that maps from observation $o$ and goal $I_g$ to the next action $a$, maximizing the sum of discounted rewards $\mathbb{E}_{s_0 \sim \rho_0, I_g \sim \mathcal{G}} \sum_t \gamma^t \mathcal{R}(s_t, I_g)$. Our policy $\pi(a|o, I_g)$ is a deterministic policy $\mu_\theta$ parameterized by $\theta$, outputting continuous actions $a_t = \mu_\theta(o_t, I_g)$.

### 3.3 Photo Finishing State Representation

State presentation is also critical for the success of the proposed RL policy. This is particularly important in challenging scenarios where the policy must tune parameters for unseen goals of any style. Our experiment in Sec. 4.3 also shows that a simple concatenation of the input and goal images

yields a sub-optimal result. Therefore, we design a comprehensive photo finishing state representation to extract features from observations that are critical for photo finishing tuning.

Specifically, our representation consists of three components: a CNN-based dual-path feature representation to encode both global and local features, a photo statistics representation to match the traditional photographic statistics between input and goal images, and an embedding of historical actions. Details of each component are described below.

**Dual-path feature representation.** In our dual-path feature representation, we seek to extract both global and local features from both input and goal images, inspired by [8]. This is because the tuning task requires not only global image characteristics such as overall color, tone, average intensity, and scene category, but also local features associated with texture, highlight, and shadow. As shown in Fig. 2, the architecture begins with a stride-2 convolutional encoder to reduce spatial resolution and extract initial low-level features. It then splits into two paths: a local path and a global path. The local path $\{L^i\}_{i=1...N_L}$ includes two stride-1 convolutional layers, preserving spatial resolution to extract local features. The global path $\{G^i\}_{i=1...N_G}$ has one stride-2 convolutional layer and three fully-connected layers, providing a global scene summary vector. This global feature encapsulates essential global image characteristics such as overall color and tone, as well as a global notion of scene category (light condition or indoor/outdoor). We fuse the global and local features by adding the global feature at each $x, y$ spatial location of the local feature: $F^D_{x,y} = \sigma(G^{N_G} + L^{N_L}_{x,y})$, where $\sigma$ is the ReLU activation. This results in a dual-path feature representation $F^D$.

**Photo statistics representation.** Simply relying on a CNN-based policy may lead to unsatisfactory results with input and goal images outside of training distribution. To better represent invariant features across diverse styles and content of both input and goal images, we introduce a photo statistics representation, which matches traditional image statistics such as a histogram between input and goal. Global photo statistics, such as histograms, are critical in global image processing operations, such as exposure control [22], highlight, and shadow. Since they cannot be well represented by conventional convolutional neural networks due to limited receptive fields [29], we propose to pre-compute these statistics and concatenate them into our state representation. Specifically, we compute histograms $H_{rgb}$ on the RGB channels of input and goal images and map these to a fixed dimension using a linear layer $H' = \text{Linear}((H_{rgb}(I_t); H_{rgb}(I_g))$. This feature is then concatenated with the luminance, median, contrast, and saturation of both input and goal images to form the photo statistics representation $F^S$.

**Policy network.** At last, we combine the dual-path feature representation and photo statistics representation with a historical action embedding $F^H_t = \text{Linear}(a_{1:t}; \ell_{1:t})$, where $\ell_t$ is the $\ell_2$-distance of image $I_t$ and goal image $I_g$. As shown in Fig. 2, the input of policy network can be formulated as $s_t = \text{Concat}(F^D_t; F^S_t; F^H_t)$. The policy network is a multi-layer perceptron network (MLP) that maps from the current state and the goal representation to the next action to take. We choose deterministic policy to directly output continuous action $a_t = \mu_\theta(o_t, g) = \mu_\theta(s_t)$. We use the same architecture to estimate the value function for RL updates.

### 3.4 Reward Function and Training Objectives

We provide a general RL-based framework for photo tuning tasks. One can train our goal-conditioned end-to-end policy with different reward functions to resolve different photo tuning tasks including photo finishing tuning and photo stylization tuning. The policy is optimized with twin-delayed DDPG (TD3) algorithm [5].

**Reward function for photo finishing tuning.** When goal image $I_g$ is the photo-finished input image, we measure the distance between current image $I_t$ and $I_g$ with PSNR metric. The reward function is calculated as the difference between PSNR values of consecutive steps:

$$r_t = \text{PSNR}(I_{t+1}, I_g) - \text{PSNR}(I_t, I_g). \tag{2}$$

Instead of using $\ell_2$-distance to measure image distance, we use PSNR, the negative logarithm of $\ell_2$-distance. This design ensures the policy receives appropriate rewards even when the current image is close to the goal, encouraging fine-grained tuning of pipeline parameters.

**Reward function for photo stylization tuning.** When goal image $I_g$ is a style image with arbitrary content and style, we measure the distance between the input and goal images with a style score:

$$\text{StyleScore}_t = \sum_{i=1}^{N_S} \|G_i[I_g] - G_i[I_t]\|_2 + \lambda_0 \|H(I_t^Y), H(I_g^Y)\|_2 + \lambda_1 \|H(I_t^{UV}), H(I_g^{UV})\|_2, \tag{3}$$

where $N_S$ denotes the number of layers from a pre-trained VGG-19 [27] model used to extract features. Following [6], the style is captured using Gram matrices $G_i[\cdot] = F_i[\cdot]F_i[\cdot]^T$, with $F_i$ representing feature maps. Additionally, the $\ell_2$-distances of histograms for the Y and UV channels are included to align luminance and color palettes. The overall reward is calculated by the change in style score across consecutive steps, penalized by the difference in content features $\bar{F}_i$:

$$r_t = \text{StyleScore}_t - \text{StyleScore}_{t+1} - \lambda_2 \sum_{i=1}^{N_C} \|F_i[I_g] - F_i[I_t]\|_2 . \tag{4}$$

**Policy optimization.** We optimize our goal-conditioned policy with off-policy TD3 [5] algorithm. Specifically, TD3 learns two Q value function $Q_{\phi_1}$ and $Q_{\phi_2}$, optimized by mean square Bellman error minimization:

$$y(r_t, s_{t+1}, d) = r_t + \gamma(1-d) \min_{i=1,2} Q_{\phi_{i,\text{targ}}}(s_{t+1}, a'(s_{t+1})), \tag{5}$$

$$L(\phi_i) = \mathop{\mathbb{E}}_{s_t \sim \mathcal{D}} \left[ (Q_{\phi_i}(s_t, a_t) - y(r_t, s_{t+1}, d))^2 \right], \tag{6}$$

where $Q_{\phi_{i,\text{targ}}}$ is the exponential moving average of $Q_{\phi_i}$, $a'(s_{t+1})$ is given by target policy with clipped gaussian noise, and $d$ is the termination signal. The state transition pair $(s_t, a_t, r_t, s_{t+1}, d)$ is sampled from a replay buffer $\mathcal{D}$. With Q functions, the policy $\mu_\theta(\cdot)$ is learned by maximizing $Q_{\phi_1}$:

$$L(\theta) = \mathop{\mathbb{E}}_{s_t \sim \mathcal{D}} [Q_{\phi_1}(s_t, \mu_\theta(s_t))] . \tag{7}$$

More details about our reward functions and policy optimization are in the appendix.

## 4 Experiments

In the first subsection, we provide the details of datasets and task settings for photo finishing tuning and photo stylization tuning, along with a description of the evaluation metrics. In the second subsection, we demonstrate our experimental results and compare them to zeroth-order optimization [20] and first-order optimization [28, 29] baseline methods. Ablation studies are conducted in the last subsection, which investigates the impact of each component of our state representation. We also provide supplementary qualitative results in the Appendix.

### 4.1 Tasks Settings and Datasets

**Datasets.** We use the MIT-Adobe FiveK Dataset [3], a renowned resource in the field of photo retouching, which comprises 5,000 photographs captured using DSLR cameras by various photographers. This dataset is notable for providing images in raw format alongside the retouching outcomes of five experts. For our study, we selected 4,500 images to serve as the training dataset, with the remaining 500 images designated as the validation dataset. In our method, random parameters are employed to generate the target images, which are used as training data pairs. In the task of photo finishing tuning, the datasets including both the expert C retouched targets and randomly generated targets are utilized for evaluation. The expert C retouched targets are optimized using CMA-ES to ensure they are reachable by our image processing pipeline. For the photo stylization tuning task, we have curated a collection of 200 diverse style images from the Lightroom Discover website [3], following [26].

To further evaluate our method and demonstrate its generalizability, we test our RL-based framework directly on the HDR+ dataset [10]. We used the official subset of the HDR+ dataset, which consists of 153 scenes, each containing up to 10 raw photos. The aligned and merged frames are used as the input, expertly tuned images serve as the photo-finishing targets.

**Implementation details.** We conduct all experiments on an image processing pipeline consisting of standard image processing operations, including exposure, color balance, saturation, contrast, tone mapping (highlight and shadow), and texture (sharpness and smoothing), with nine adjustable parameters in total, similar to [29]. Thus, the agent's action space is comprised of fine continuous actions corresponding to these nine pipeline parameters. During the policy inference, the input and goal images are resized to the resolution of $64 \times 64$. Additionally, a 3-level Laplacian pyramid of

---

[3]https://lightroom.adobe.com/learn/discover

Table 1: photo finishing tuning experimental results on the FiveK validation datasets with FiveK targets (expert-C) and random targets. Queries represent the times of query image processing pipeline.

| Eval Dataset | FiveK-Target | | | | Random-Target | | | |
|---|---|---|---|---|---|---|---|---|
| Method | PSNR↑ | SSIM↑ | LPIPS↓ | Queries↓ | PSNR↑ | SSIM↑ | LPIPS↓ | Queries↓ |
| CMAES [9, 20] | 28.53 | 0.9586 | 0.0968 | 200 | 32.29 | 0.9754 | 0.0827 | 200 |
| Monolithic Proxy [28] | 21.71 | 0.9104 | 0.2144 | - | 21.08 | 0.9251 | 0.2785 | - |
| Cascaded Proxy [29] | 22.31 | 0.9115 | 0.1939 | - | 21.40 | 0.9213 | 0.2613 | - |
| **Ours** | **35.89** | **0.9764** | **0.0305** | **10** | **38.46** | **0.9814** | **0.0128** | **10** |

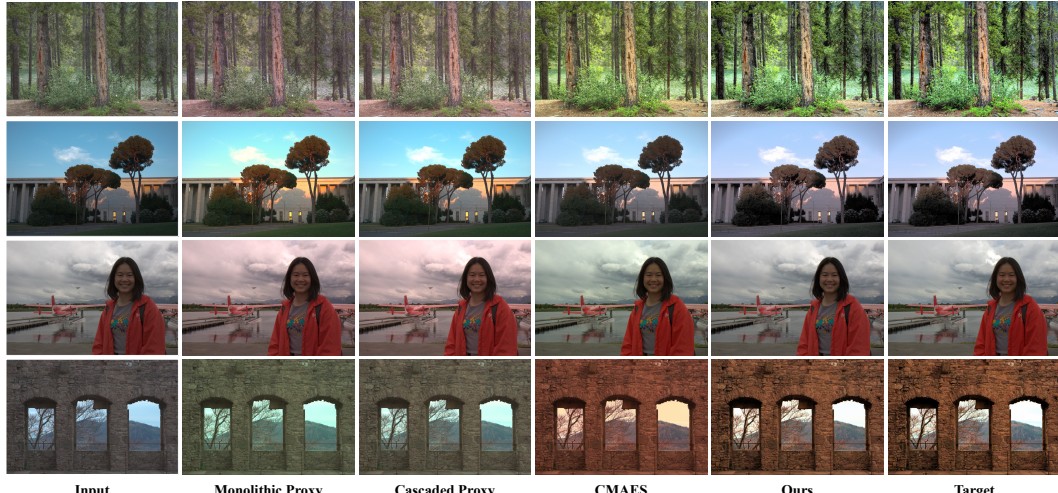

Input    Monolithic Proxy    Cascaded Proxy    CMAES    Ours    Target

Figure 3: Photo finishing tuning results on FiveK dataset with expert C target. The visual results of our method are closest to the target image, especially in terms of color and brightness.

both input and goal images is constructed and fed into the policy network to capture high-frequency details from the original resolution. Then the output parameters from the policy network are fed to the image processing pipeline along with full-resolution images to produce high-resolution results. We train our policy using the standard TD3 algorithm [5] and set the termination of our RL policy to trigger when the episode length reaches the maximum threshold (10 steps), ensuring efficiency. Further details can be found in the appendix.

**Evaluate metrics.** Similar to [11, 28, 29], we employ the Peak Signal-to-Noise Ratio (PSNR), the Structural Similarity Index Measure (SSIM) [30], and the Learned Perceptual Image Patch Similarity (LPIPS) [33] as our evaluation metrics. To assess the quality of stylization, we conduct user studies to evaluate our methods, offering a subjective measure of image quality based on viewer assessments.

### 4.2 Results

**Results of photo finishing tuning.** To evaluate the efficacy of our framework on the photo finishing tuning Task, we conduct two experiments on the FiveK validation datasets, using both expert-C targets and random targets. Our method is compared against the monolithic proxy-based approach [28], the cascaded proxy-based method [29], and the search-based method proposed [20]. Consistent with [29], we utilize 100 iterations during inference for the proxy-based methods. Additionally, we record the number of times that the photo-finishing pipeline was queried in both search-based methods and our approach.

As illustrated in Tab. 1, our method outperforms the others across all metrics on both FiveK-Target and Random-Target. The monolithic proxy-based method struggles with accurately representing the complex image processing pipeline, leading to suboptimal performance. While the cascaded proxy method can incrementally enhance tuning performance over its monolithic counterpart, it suffers from accumulated errors and difficulties in approximating certain operations, such as texture, resulting in poorer performance. Notably, our method only requires querying 10 times image processing pipeline,

Table 2: Experimental results demonstrating efficiency across varying input resolutions. Our method significantly outperforms other methods, achieving a speed enhancement of 260 times relative to the cascaded proxy method [29] at 720P resolution, and 117 times faster than the CMAES approach [9, 20] at 4K resolution.

| Methods | Monolithic Proxy [28] | | | | Cascaded Proxy [29] | | | | CMAES [9, 20] | | | | Ours | | | |
|---|---|---|---|---|---|---|---|---|---|---|---|---|---|---|---|---|
| Resolution | 720P | 1K | 2K | 4K | 720P | 1K | 2K | 4K | 720P | 1K | 2K | 4K | 720P | 1K | 2K | 4K |
| Time(s) | 7.67 | 17.89 | 33.07 | OOM | 70.45 | OOM | OOM | OOM | 36.16 | 51.82 | 91.86 | 144.02 | **0.27** | **0.33** | **0.47** | **1.23** |

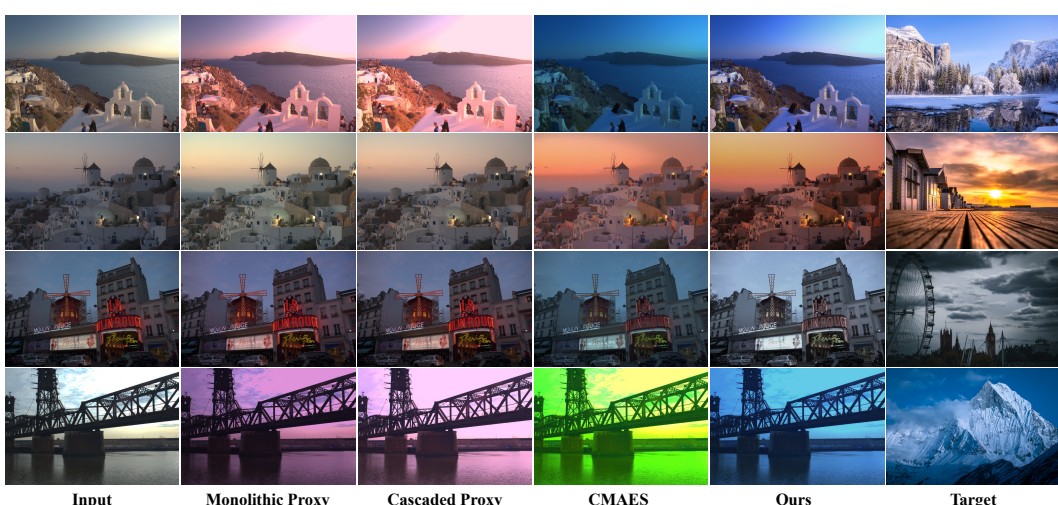

| Input | Monolithic Proxy | Cascaded Proxy | CMAES | Ours | Target |

Figure 4: Photo stylization tuning results. Compared with CMAES [9, 20], monolithic proxy [28], and cascaded proxy [29], our output matches the best with the style goal.

whereas the performance of the search-based method, even with 200 queries, remained inferior to ours. Further, the visualization results depicted in Fig. 4 demonstrate that our method produces visual outcomes that most closely match the target images, particularly in terms of color and brightness, when compared to all other methods. More visualization can be found in Fig. 7 of Appendix A.1.

**Efficiency.** To evaluate the efficiency of our approach, we conducted speed testing experiments on a system equipped with an AMD EPYC 7402 (48C) @ 2.8 GHz CPU, 8 NVIDIA RTX 4090 GPUs with 24GB of RAM each, 512 GB of memory, and running CentOS 7.9. In line with [29], we applied 200 iterations during inference for both the proxy-based methods [28, 29] and the search-based method [20]. We measured the execution time for each method across four different input resolutions.

As indicated in Tab. 2, our method demonstrated superior efficiency, requiring only 1.23 seconds per execution with 4K input. While the monolithic proxy-based method outperformed the search-based method in terms of speed, it faced limitations as input resolutions increased, leading to out-of-memory (OOM) errors once GPU memory was exceeded. The cascaded proxy-based method, which includes MLP networks, was the slowest and most prone to memory overflows due to its intensive memory demands. The search-based method primarily depends on CPU performance. Despite being executed on a high-performance server, the CMAES method [20] requires 144 seconds to process a 4K input image, which is considerably slow.

**Results of photo stylization tuning.** We conduct qualitative comparisons and user studies on the photo stylization tuning task to demonstrate the effectiveness of our goal-conditioned RL framework. Our policy was trained using the FiveK training dataset, and all methods were evaluated on input and style image pairs collected from the Adobe Lightroom Discover website. For all baseline methods, we adopt the same style score described Sec.3.4 as optimization objectives, ensuring fair comparison.

As shown in Fig. 4, our method produces results closer to the style goal image compared to the baseline. Notably, our method achieves better results with only 10 queries to the image processing pipeline, whereas the baseline methods require 200 queries, making them orders of magnitude slower. It is important to note that the style images from the Lightroom Discover website have a different distribution than our training dataset. Despite this, our method adapts directly to the target style image distribution during testing, whereas the baseline methods require time-consuming optimization of testing data. These experimental results demonstrate that our RL-based framework can efficiently tune

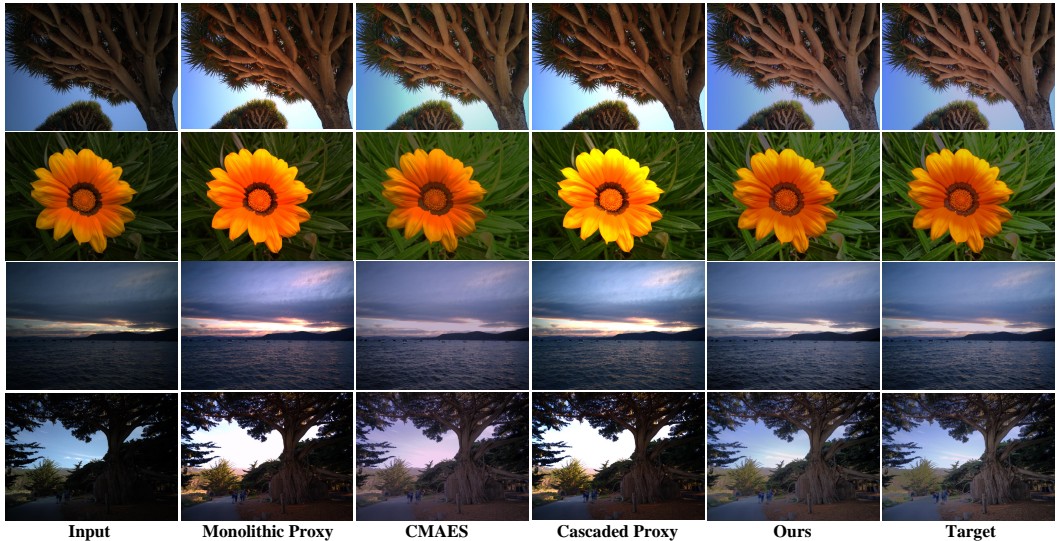

| Input | Monolithic Proxy | CMAES | Cascaded Proxy | Ours | Target |

Figure 5: Qualitative comparison on the HDR+ photo finishing tuning task. These comparisons illustrate that our method remains closer to the target even when dealing with input and target images outside the training distribution.

Table 3: Photo finishing tuning experimental results on addition HDR+ datasets [10] with HDR+ expert-tuned targets.

| Eval Dataset | HDR+ Target | | | |
|---|---|---|---|---|
| Method | PSNR↑ | SSIM↑ | LPIPS↓ | Queries↓ |
| CMAES [9, 20] | 28.08 | 0.9539 | 0.1307 | 200 |
| Greedy Search [16] | 25.79 | 0.9212 | 0.1542 | 200 |
| Monolithic Proxy [28] | 17.80 | 0.8940 | 0.3044 | - |
| Cascaded Proxy [29] | 18.90 | 0.8982 | 0.2797 | - |
| **Ours** | **31.54** | **0.9652** | **0.0563** | **10** |

images to different unseen styles, showcasing that our photo finishing state representation (Sec. 3.3) has the capability to generalize to versatile goals outside of training distributions. More visualization with versatile goal images can be found in Fig. 8 of Appendix A.2.

To rigorously evaluate the effectiveness of our method in photo stylization tuning, we implemented a subjective user study comprising 20 questions. In each question, participants were presented with images generated by four different methods—monolithic proxy, cascaded proxy, CMAES, and our own approach. Participants were asked to identify up to two images that most closely resembled a given target image. The study was conducted online, garnering 65 responses from a diverse group of individuals selected randomly from the internet.

The aggregated preferences are visually summarized in Fig. 6. The data clearly show that our method is perceived by the majority of participants as producing results that most closely match the target images, highlighting its superiority in stylization tuning tasks.

**Cross dataset generalization.** We conducted additional evaluations using the HDR+ dataset [10] to demonstrate our RL-based framework's ability to generalize effectively to unseen datasets. We compare to baselines including CMAES [9, 20], Cascaded Proxy [29], Monolithic Proxy [28], and Greedy Search [16]. In Tab. 3, we report PSNR, SSIM, LPIPS, and queries to the ISP pipeline. The results demonstrate that our RL policy generalizes effectively to unseen data, achieving higher photo-finishing quality than methods directly tuned on the test dataset. Qualitative comparisons in Fig. 5 show that our results are closer to targets, even with input and target images outside the training distribution.

In Tab. 3, our RL-based method achieves a PSNR of 31.54 on the HDR+ photo-finishing task, and it outperforms all baselines. This shows that our RL policy, trained on the FiveK dataset, effectively generalizes to the HDR+ dataset. Such out-of-distribution capability is facilitated by our proposed

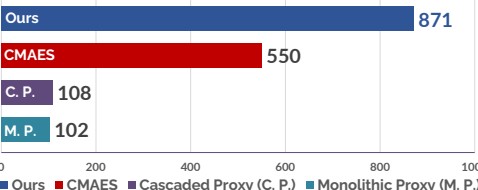

Figure 6: Results of the user study on the photo stylization tuning task. Each bar represents the number of votes each method received, where participants select the images they believed most closely resembled target images.

Table 4: Ablation study on each component of our finishing state representation. $RL$ denotes a baseline naively using CNN trained with RL, $F^D, F^S, F^H$ denotes each of our state representation respectively.

|  | $RL$ | $F^D$ | $F^S$ | $F^H$ | PSNR↑ | SSIM↑ |
|---|---|---|---|---|---|---|
| $Ex_1$ | ✓ |  |  |  | 32.17 | 0.968 |
| $Ex_2$ | ✓ | ✓ |  |  | 35.15 | 0.973 |
| $Ex_3$ | ✓ | ✓ | ✓ |  | 37.61 | 0.980 |
| $Ex_4$ | ✓ | ✓ |  | ✓ | 35.96 | 0.976 |
| $Ex_5$ | ✓ | ✓ | ✓ | ✓ | 38.46 | 0.983 |

photo-finishing state representation, which extracts invariant features for photo finishing, allowing adaptation to diverse inputs and goals beyond the training distributions. The CMAES [9, 20] baseline shows consistent results on HDR+ compared to FiveK, as it is directly optimized on the test dataset without prior training. However, proxy-based methods [28, 29] perform worse because the proxy network trained on FiveK does not generalize well to HDR+, leading to incorrect gradients and poorer photo-finishing quality.

### 4.3 Ablation Study on State Representation

As has been shown in our main experiment, our RL-based approach significantly outperforms the previous method in terms of photo finishing quality and efficiency, demonstrating that RL is more suitable for the photo finishing tuning task. In this subsection, we focus on the photo finishing state representation we propose to better fit RL in our task. Specifically, we conduct experiments on photo finishing tuning task using FiveK Random-Target dataset, in order to study the contribution of each specific representation in our photo finishing state representation.

As shown in Tab. 4, we set our baseline $RL$ as RL policy trained with a naive CNN-based encoder taking the concatenated input and goal images as input. This baseline achieves only 32.17 dB of PSNR. In $Ex_2$, our dual-path feature representation $F^D$ improves the photo finishing quality, as it better encodes local features and global image characteristics critical to our photo tuning task. Since traditional photo statistics contain invariant features about global statistics that are difficult to learn solely using a network, the photo statistics representation $F^S$ also helps to boost photo finishing quality as shown in $Ex_3$. Moreover, as parameters and results from all previous RL steps help in the decision process, the historical action representation is also useful as shown in $Ex_4$. With the proposed three representations combined, our RL policy can be guided to effectively tune image processing pipeline parameters given input and goal images of any photo finishing style, as evidenced by the superior performance in $Ex_5$.

## 5 Conclusion

In this work, we propose an RL-based photo finishing tuning algorithm that efficiently tunes the parameters of a black-box image processing pipeline to match any tuning target. Our approach encompasses several key innovations. Firstly, we integrate goal-conditioned RL into the realm of photo-finishing tuning. Secondly, we propose a photo finishing state representation comprising three principal components essential for training an effective RL policy network: a CNN-based feature representation that encodes both global and local image features, a photo statistics representation designed to align the photographic statistics between the input and the target, and an embedding of historical actions. We assess the effectiveness of our framework through comprehensive experiments on both photo finishing tuning and photo stylization tuning. The experimental results affirm that our goal-conditioned policy is an adept photo-finishing tuner, capable of exerting efficient and fine-grained control over image processing pipeline parameters to fulfill a variety of objectives. Currently, our method exclusively supports conditional inputs in the form of images and lacks the capability to process non-image types such as textual inputs. Moving forward, we intend to broaden our research to include multi-modal conditional inputs.

## Acknowledgements

This work is supported by Shanghai Artificial Intelligence Laboratory and RGC Early Career Scheme (ECS) No. 24209224. We also extend our gratitude to Quanyi Li for his insightful discussions and valuable comments.

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

# A Additional Results

## A.1 Photo Finishing Tuning Results on FiveK Dataset

We show more photo finishing tuning visualization results on the FiveK-Target evaluation dataset in Fig. 7, showcasing its superior performance in terms of color and brightness compared to all other methods.

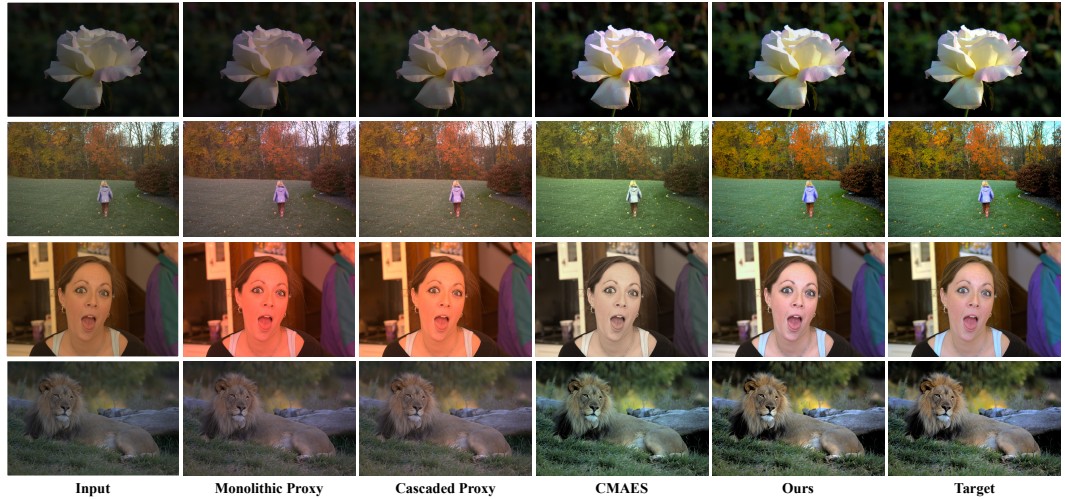

Input     Monolithic Proxy     Cascaded Proxy     CMAES     Ours     Target

Figure 7: Additional results on photo finishing tuning task.

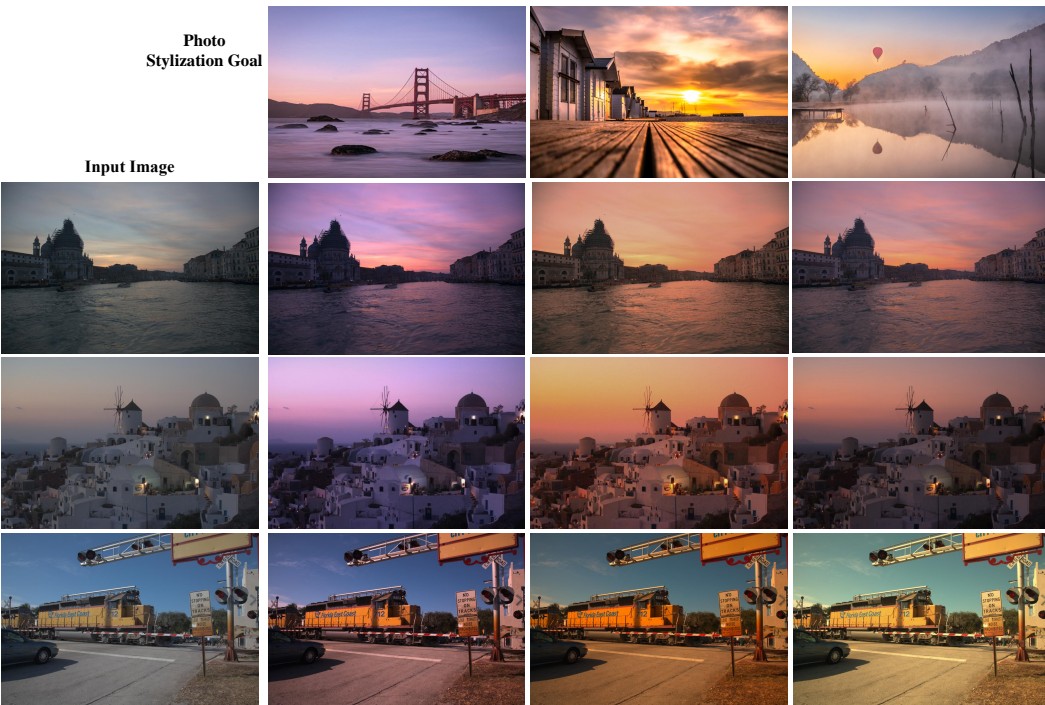

Figure 8: Additional results on photo stylization tuning task.

## A.2 Photo Stylization Tuning Results

We show more photo stylization tuning visualization results in Fig. 8, showcasing its superior performance in tuning input images to any style goal that is outside the training goal distribution.

# B Image Processing Pipeline

The detailed image processing pipeline in this paper encompasses various standard operations, including:

(1) Exposure: Adjusts the overall brightness of the image, primarily affecting the amount of light captured. Increasing exposure makes the image brighter while decreasing it makes the image darker. This is useful for correcting overexposed or underexposed photos. This operation includes one slider parameter.

(2) Color Balance: Used to adjust the color temperature and tint of an image, altering its overall color feel. Adjusting the color temperature (blue to yellow) and tint (green to magenta) can make the image appear warmer or cooler. This operation includes three slider parameters.

(3) Saturation: Controls the intensity of colors in the image. Increasing saturation makes the colors more vivid and lively, whereas decreasing saturation reduces color intensity, making the image appear softer or closer to black and white. This operation includes one slider parameter.

(4) Contrast: Adjusts the difference between the brightest and darkest parts of the image. Increasing contrast makes dark areas darker and bright areas brighter, enhancing the depth and dimension of the image. Decreasing contrast brings these areas closer to mid-grey, reducing the image's depth. This operation includes one slider parameter.

(5) Tone Mapping: This operation compresses a high dynamic range input to a smaller dynamic range, which is affected by the Highlights and Shadows sliders, respectively. Highlights: Adjusts the brightness of the brightest areas of the image without affecting overall exposure. This helps recover details in overexposed areas. Shadows: Adjusts the brightness of the darkest areas, helping reveal details in shadowed regions without changing the overall exposure. This operation includes two slider parameters.

(6) Texture: Enhances or reduces the detail and texture in the image without affecting colors. Enhancing texture can make details in the image clearer, while reducing texture can smooth out the image, often used in portrait photos for skin treatment. This operation includes one slider parameter.

# C Additional Implementation Details and Training Strategy

## C.1 Training Details of TD3 Algorithm

We state the details of TD3 [5] algorithm training in this subsection. TD3 is an off-policy RL algorithm. When the policy sample transition pair $(s_t, a_t, r_t, s_{t+1}, d)$ to form the replay buffer $\mathcal{D}$, a gaussian noise is added to encourage exploration: $a = \text{clip}\left(\mu_\theta(s) + \epsilon, a_{Low}, a_{High}\right)$, where $\epsilon \sim \mathcal{N}(0, \sigma)$. We set $\sigma = 0.1$ for a trade-off between exploration and exploitation.

To compute the target action in Q update target:

$$L(\phi_i) = \mathop{\mathbb{E}}_{s_t \sim \mathcal{D}} \left[ \left( Q_{\phi_i}(s_t, a_t) - \left( r_t + \gamma(1 - d) \min_{i=1,2} Q_{\phi_{i,\text{targ}}}(s_{t+1}, a'(s_{t+1})) \right) \right)^2 \right], \qquad (8)$$

The next action of $s_{t+1}$ comes from the target policy with a clipped noise, so that incorrect action peak produced by sub-optimal Q value estimation is smoothed out, as in the following equation:

$$a'(s_{t+1}) = \text{clip}\left( \mu_{\theta_{\text{targ}}}(s_{t+1}) + \text{clip}(\epsilon_1, -c, c), a_{Low}, a_{High} \right), \qquad \epsilon_1 \sim \mathcal{N}(0, \sigma) \qquad (9)$$

In this implementation, we set $\epsilon_1 = 0.2$, which is twice the value of $\epsilon$. TD3 updates the policy less frequently than the Q-function. We set the policy to update half as frequently as the Q network update. The optimization objectives of the policy are already given in the main paper. To ensure a robust

update of the policy and Q network, TD3 adopts an Exponential Moving Average (EMA) strategy to update the target network Q and policy network which is used to compute the Q value target above.

$$\phi_{\text{targ},1} \leftarrow \rho\phi_{\text{targ},1} + (1 - \rho)\phi, \tag{10}$$

$$\phi_{\text{targ},2} \leftarrow \rho\phi_{\text{targ},2} + (1 - \rho)\phi, \tag{11}$$

$$\theta_{\text{targ}} \leftarrow \rho\theta_{\text{targ}} + (1 - \rho)\theta \tag{12}$$

We set the EMA update rate $\rho = 0.99$ in all our experiments. The optimizer is performed using Adam optimizer [17] with $(\beta_1, \beta_2) = (0.9, 0.999)$. We set the learning rate to specific values for policy and value network, that is, 1e-4 for action and 2e-4 for Q network. We set batch size as 64 for both photo finishing tuning and photo stylization tuning experiments. We set the discount factor $\gamma = 0.9$. All experiments are conducted on NVIDIA RTX 4090 GPUs. Furthermore, we implement the CMAES method [28] based on open-source framework [1].

### C.2 Other Training Details

**Reward design.** For our detailed reward design, as described in Sec. 3.4, we utilize the PSNR metric of consecutive steps for photo finishing tuning. In photo stylization tuning, we adopt the style score as the main reward and add a content negative reward to prevent the policy from taking drastic actions that hinder photo stylization quality. Specifically, we set $\lambda_0$ and $\lambda_1$ in StyleScore$_t$ to 100 and 50, respectively, and set $\lambda_2 = 0.5$. For the VGG features selected to compute the style score, we follow [12] to set $N_S = N_C = 4$, selecting relu_{1...4}_1 features to form $\{F_i\}_{i=1...4}$, which is then used to compute then gram matrix and the content regularization term.

**RL termination.** Since our RL policy freely explores the image processing pipeline parameter space, we need to terminate the current episode if the state collapses, meaning the policy outputs parameters that render an abnormal image. Specifically, we calculate the average pixel intensity of the image $I_t$ as $\mathcal{I}_t$ and set this value to be within the range of $\mathcal{I}_{min}$ and $\mathcal{I}_{max}$. For each rollout, if $\mathcal{I}_t$ is out of range, we terminate the state and do not save it to our replay buffer $\mathcal{D}$.

**Network architecture.** Our approach learns an end-to-end policy that maps from input images and goal specifications to the next action to take (the next parameters of the image processing pipeline.) The state representation is $s_t$ as described in Sec. 3.3, which is fed into a 4-layer multi-layer perceptron network (MLP), with a continuous action space of 9 parameters. Each hidden layer of the MLP network is of width 512.

### C.3 Baselines Implementation Details

**Implementation details for CMAES [9, 20].** CMAES is a gradient-free search (zeroth order optimization) method using an evolution strategy. As [20] does not provide code, we implemented CMAES using the pymoo library [1], enabling parallel execution on multi-core CPUs and achieving reasonable performance. This baseline does not require training.

**Implementation details for Cascaded Proxy [29].** For proxy network training, we followed the architecture in [29], using 3 consecutive $1 \times 1$ convolutions for pointwise ISP operations and 5 consecutive $3 \times 3$ convolutions for areawise operations. We trained with the Adam optimizer, a learning rate of 1e-4, a batch size of 512, and 100 epochs, as recommended. Camera metadata was extracted from DNG files, as described in [29]. Since [29] does not release its dataset, we used the MIT-Adobe FiveK dataset, as in our method. Following Section 5 of [29], we used 1,000 raw images from FiveK and sampled 100 points for each ISP hyperparameter.

**Implementation details for Monolithic Proxy [28].** The architecture uses a single UNet to approximate the ISP pipeline, with hyperparameters conditioned by concatenating extra planes to the features. We trained the proxy with the Adam optimizer, a learning rate of 1e-4, a batch size of 512, and 100 epochs. The training set generation and first-order optimization method are the same as for [29].

## D  Detailed Content of the User Study

We conduct a human subject study on the user preference among results of methods for photo stylization tuning given target style images. In this appendix section, we provide the original questions that we used to ask for users' feedback.

For each of the 20 questions in our user study, the order of the methods was randomized and labeled as A, B, C, and D to ensure the anonymity of the techniques used. Participants were not aware of which method corresponded to each label, eliminating any potential bias in their selections. The prompt for all questions remained consistent: "Among the four images on the right (labeled A, B, C, and D), which one(s) most closely resembles the target image on the left? You may select up to two images if you believe they are equally similar to the target image. Otherwise, please choose only one."

Detailed below are the options provided for each of the 20 questions.

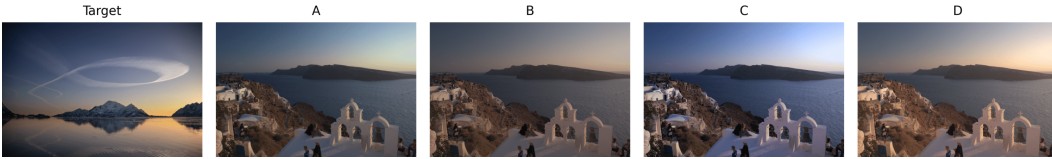

Figure 9: Options in question 1 of our user study.

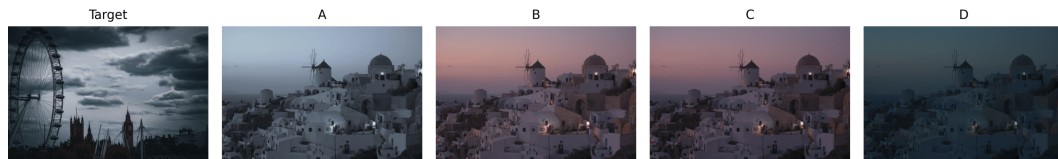

Figure 10: Options in question 2 of our user study.

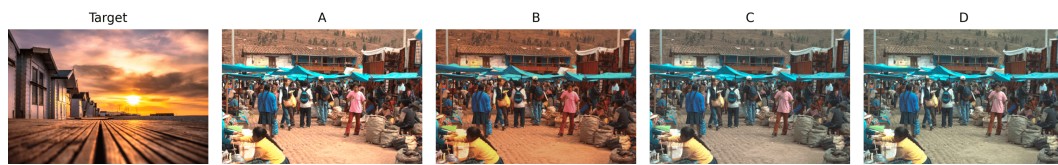

Figure 11: Options in question 3 of our user study.

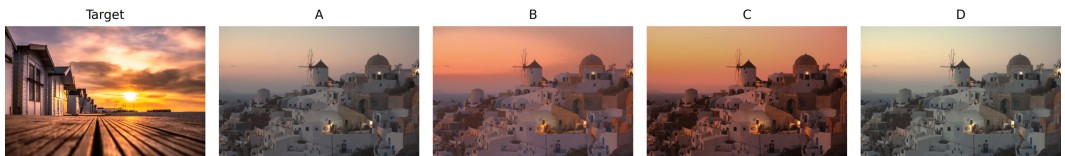

Figure 12: Options in question 4 of our user study.

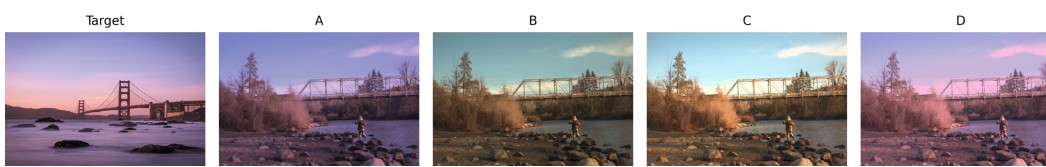

Figure 13: Options in question 5 of our user study.

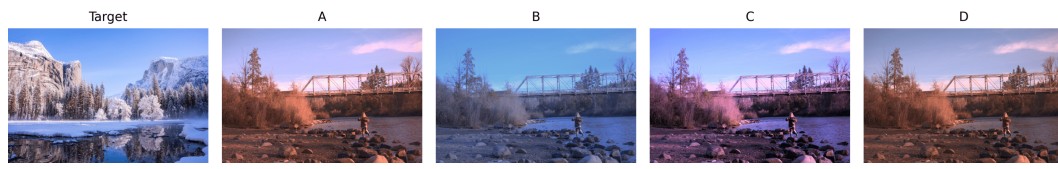

Figure 14: Options in question 6 of our user study.

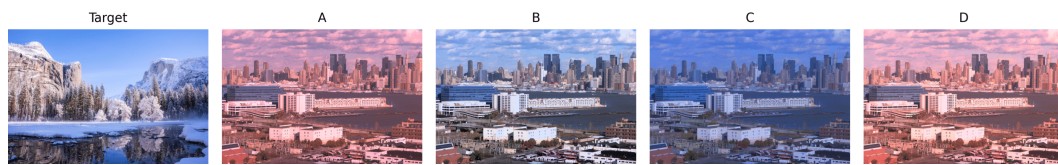

Figure 15: Options in question 7 of our user study.

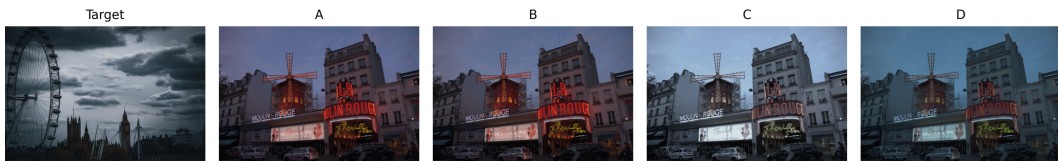

Figure 16: Options in question 8 of our user study.

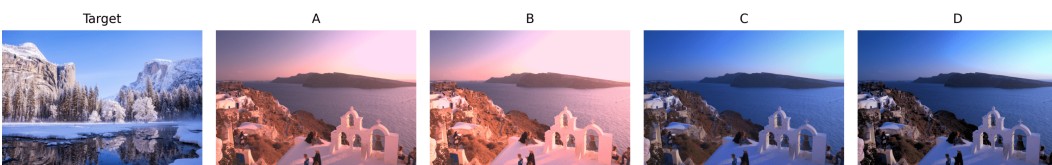

Figure 17: Options in question 9 of our user study.

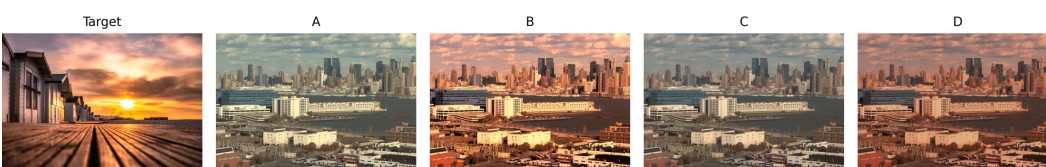

Figure 18: Options in question 10 of our user study.

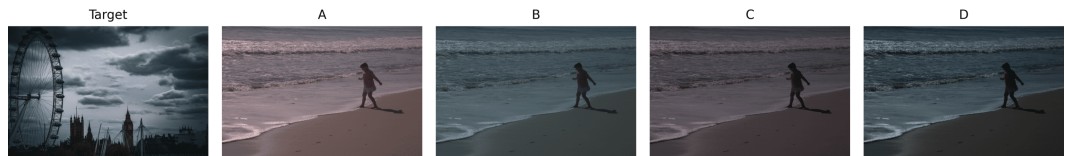

Figure 19: Options in question 11 of our user study.

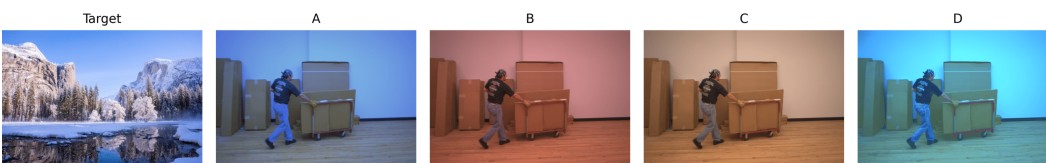

Figure 20: Options in question 12 of our user study.

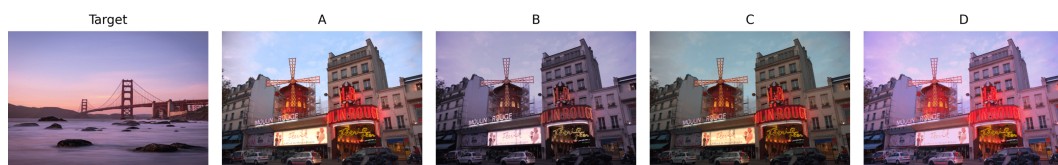

Figure 21: Options in question 13 of our user study.

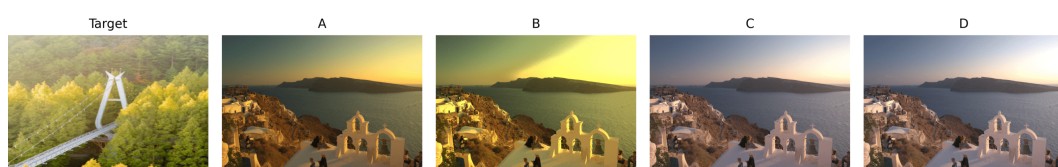

Figure 22: Options in question 14 of our user study.

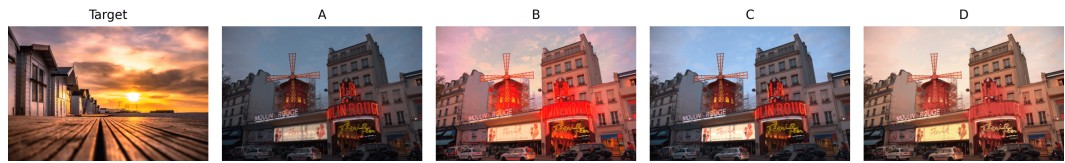

Figure 23: Options in question 15 of our user study.

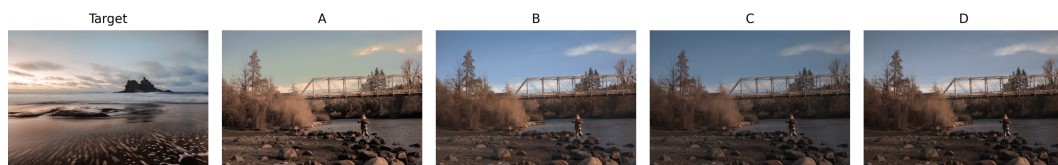

Figure 24: Options in question 16 of our user study.

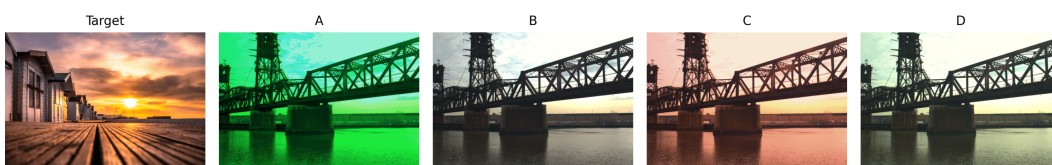

Figure 25: Options in question 17 of our user study.

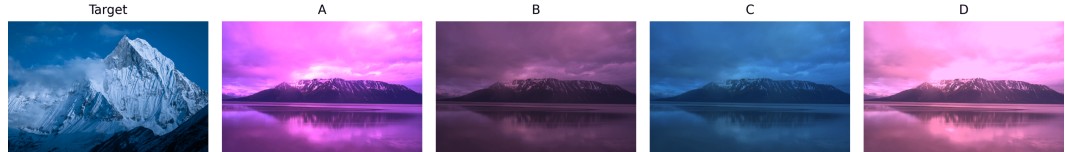

Figure 26: Options in question 18 of our user study.

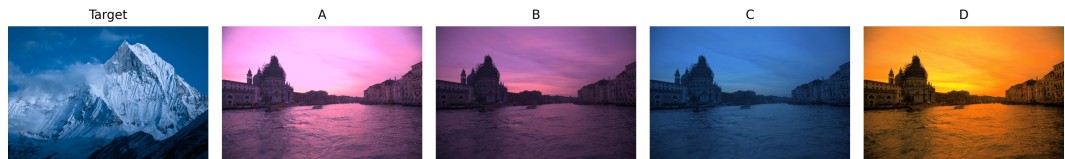

Figure 27: Options in question 19 of our user study.

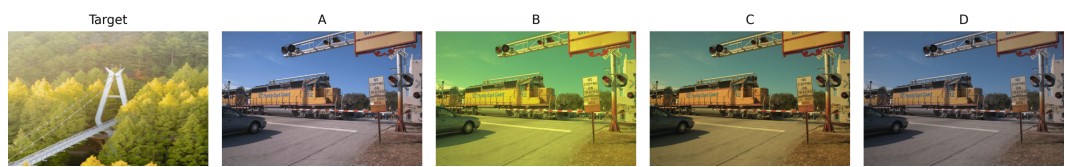

Figure 28: Options in question 20 of our user study.

