# OpenReview forum: "Goal Conditioned Reinforcement Learning for Photo Finishing Tuning"
_NeurIPS.cc/2024/Conference — NeurIPS 2024 poster_

### Official Review · Reviewer_WirW · 2024-07-09

**Soundness:** 3
**Presentation:** 3
**Contribution:** 2
**Rating:** 6
**Confidence:** 3

**Summary:**

This paper proposes a goal-conditioned reinforcement learning framework for photo finishing tuning. They introduce a novel state representation and treat the image processing pipeline as a black box, avoiding the need for differentiable proxies. The method can efficiently tune parameters to match various goals, including pixel-aligned target images and style images.

**Strengths:**

1. The paper is well-written and easy to follow, clearly explaining the goal-conditioned reinforcement learning approach for photo finishing tuning.
2. The visualization in Figure 1 effectively demonstrates the superiority of the proposed method compared to existing approaches, showing rapid convergence and high-quality results.

**Weaknesses:**

1. Compared to search-based methods, this proposed RL-based framework may not generalize well to unseen datasets.
2. The efficiency comparison in Table 2 may not be entirely fair. While the CMAES method runs on CPUs as required, the paper doesn't explore potential speed-ups through multi-processing.

**Questions:**

Is the search-based method (CMAES) run on a single CPU core to obtain the inference time?

**Limitations:**

The paper's proposed method may require training a new model for each new incoming dataset. This limitation could potentially impact the framework's adaptability and efficiency when applied to diverse or frequently changing data sources.

---

> ### Author Rebuttal · Authors · 2024-08-06
>
> # To Reviewer WirW
>
> We sincerely thank you for reviewing our work. After reading your comments carefully, we summarize the following questions.
>
> ## Q1: Cross-Dataset Generalization
> ***
>
> Please refer to Q1 in the **To All** section for a detailed answer to this question. To summarize, we conducted additional evaluations using the HDR+ dataset to demonstrate our RL-based framework's ability to generalize effectively to unseen datasets. Our method achieved a PSNR of 31.54 on the HDR+ photo-finishing tuning task, which is comparable to the 32.47 achieved on the FiveK dataset, and it outperforms all baselines.
>
> This highlights our method's strong generalization capabilities across different datasets, outperforming other methods such as CMAES, Greedy Search, Cascaded Proxy, and Monolithic Proxy. Our approach leverages a robust state representation that captures invariant features for photo finishing, enabling it to adapt to diverse inputs and target outputs beyond the training distributions. These results affirm that our RL-based framework is not only effective in the scenarios it was trained on but also exhibits superior performance on datasets it has not previously encountered.
>
>
> ## Q2: Efficiency Comparison of CMAEAS
> ***
> Our implementation of CMAES takes full advantage of parallel computing capabilities on a high-performance 48-core CPU, ensuring a fair efficiency comparison for efficiency.
>
> **Parallelization Enabled by `pymoo` Library**: For the CMAES method, we implemented the baseline using the widely-used Python library `pymoo`, which supports parallelization. In our experiment, we enable the parallelization options provided by `pymoo`. This is done by setting ```classpymoo.algorithms.soo.nonconvex.cmaes.CMAES(... , parallelize=True, ...)``` and enabling `elementwise_renner` when calling the `pymoo` library.
>
> **Hardware Configuration**: Our speed testing experiments were conducted on a high-performance server-level system equipped with AMD EPYC 7402 (**48 Cores**) @ 2.8 GHz CPU, 8 NVIDIA RTX 4090 GPUs, 512 GB of RAM, and running CentOS 7.9. With this configuration, we utilized the full capabilities of the 48-core CPU to explore potential speed-ups for CMAES through multi-processing. This setup ensures that the CMAES method was not constrained by computational resources and accurately reflects the method's performance potential.
>
> **Multi-Core Execution**: To clarify, the reported inference times for the CMAES method were obtained by running the algorithm on all available 48 CPU cores, rather than a single core. By doing so, we provided a realistic comparison with other methods.

---

> ### Author Response · Authors · 2024-08-12
>
> Dear Reviewer WirW,
>
> In our rebuttal, we test our model on an unseen dataset, HDR+. Results demonstrate that our method generalizes well to unseen datasets, outperforming all baselines by a large margin. We also explained that the CMAES baseline is implemented with a multi-core CPU with parallel acceleration.
>
> We want to follow up to see if our responses address your concerns. We would be very grateful to hear additional feedback from you and will provide further clarification if needed.
>
> Thank you again for your time and effort.

---

> > ### Comment · Reviewer_WirW · 2024-08-12
> >
> > Thank you for your response. It has successfully addressed all of my minor concerns. Based on this, I am willing to upgrade my score.

---

### Official Review · Reviewer_kK87 · 2024-07-12

**Soundness:** 3
**Presentation:** 3
**Contribution:** 3
**Rating:** 5
**Confidence:** 4

**Summary:**

This paper presents a method by which RL is used to drive the optimization of ISP hyperparameters for two tasks: (1) recovering photo finishing parameters and (2) mimicking reference style photo characteristics.

**Strengths:**

Clearly written.  Effective application of RL to photo finishing and stylization.  Quantitative evaluation showing substantially improved results on photo finishing in terms of quality, number of queries, and run time.  User study on photo stylization task.  Ablation study on state representations.

**Weaknesses:**

I'm a little skeptical of whether the baselines for [26] and [27] were implemented correctly.  What I don't really understand is that [26] does not provide code, [27] does provide code, but there's no mention as to whether they used the code from [27] and why the errors in the two methods look similar (almost all examples have a hue shift in a seemingly weird direction).  I also don't see the terms "monolithic proxy" or "cascaded proxy" in either of these two references.  I think these details at least need clarification for reproducibility.

I don't really understand how CMAES was implemented.  The method is cited as being from [9][18], but then the appendix states that it was implemented from [26] using pymoo.  Some examples like the green image in Figure 4 make me skeptical that this baseline was implemented correctly.

**Questions:**

Can the authors provide full details on how the baselines were implemented?  For me, acceptance is hinges on whether I can be convinced that the baselines were implemented correctly and are indeed that bad.

**Limitations:**

seems fine

---

> ### Author Rebuttal · Authors · 2024-08-07
>
> # To Reviewer kK87
> Q1 addresses the implementation details of three baselines. Q2 Q3 explain certain behavior (hue shift, green image) of baselines. Q4 extends extra details on baseline implementation and proves our baselines were implemented correctly.
> ## Q1: Full Details on Baseline Implementations, Terms "Cascaded & Monolithic"
> ***
> Please refer to **Q3** in the **To All** section for how we implement each baseline, including code used and full details. Please also refer to **Q2** in the **To All** section for clarification of the terms "cascaded & monolithic proxy".
>
> Reviewer pointed out an **incorrect citation of CMAES** in the appendix. On line 487, we make a typo on the citation of CMAES; the correct citation should be [18], not [26]. We apologize for this misunderstanding and will correct it in our paper. Since [18] does not provide code, we reproduce it using `pymoo`.
> ## Q2: Similar Hue Shift for Proxy-Based Methods [26,27]
> ***
> Hue shift is a common artifact. As shown in Figure 2 of the PDF file, snapshots from the result figures in the [27] paper also exhibit severe hue shift. In our paper, [26] and [27] have similar hue shifts in some cases, and we believe it was a result of inherent challenges in proxy-based methods, and is due to similar errors in proxy networks.
>
> The similar inaccuracies of proxy networks are due to the similar data bias introduced during proxy training data generation. [27] utilize a uniform strategy to generate proxy training data:
>
> >"We uniformly sample 100 points for each slider when generating the data for each intermediate." --Section 5,[27]
>
> This uniform sampling strategy introduces bias because sub-ranges of ISP parameters may not distribute evenly and often have non-linear effects. And ISP operations are black-box so the parameter range cannot be considered during data sampling. For example, the white balance parameter (RGB color channel coefficients) ranges from [0.6, 1.65], where 1 represents no gain. Uniform sampling results in more data points between [1, 1.65] than [0.6, 1], potentially leading to biased training data. Since the proxy network is purely data-driven, such data bias can lead to similar inaccuracies. As a result, both proxies tend to behave similarly, especially in complex ISP operations like color adjustments.
>
> Then, both methods use the same regression optimization approach with the Adam optimizer and identical hyperparameters, leading to similar optimization dynamics. It results in similar parameter shifts, particularly in white balance adjustments, where even slight inaccuracies can cause visible hue shifts. Importantly, these shifts do not occur in all cases, as shown in Figure 3 of our paper and Figure 1 of the rebuttal PDF. In conclusion, similar hue shifts are expected due to shared conditions and challenges in proxy-based ISP tuning.
> ## Q3: Green Image in CMAES Figure 4
> ***
> The color shift in Figure 4 is an occasional failure case specific to the stylization task, not the photo finishing tuning task. It occurs because style loss is too complex for the CMAES to explore the search space, causing the optimization stuck in local minimal. In contrast, our RL-based method has superior exploration capabilities under such cases.
> ## Q4: Extra Details on Each Baseline Implementation
> ***
> We extend extra details on the performance and implementation of each baseline, to prove our baselines' correctness.
> ### Cascaded Proxy [27]
> Baseline [27] underperforms due to (1) our pipeline having more image processing operations, (2) the neural proxy cannot generalize well to unseen images, and (3) the unavailability of the dataset used in [27].
>
> **Accuracy Drop with More ISP Hyperparams**: As shown in the table below, proxy approximation accuracy and finishing quality of [27] decreases with more ISP operations. This is due to error accumulation in the proxy pipeline with more ISP operations, which leads to inaccurate gradient and suboptimal parameter recovery, causing the finishing quality to decrease.
>
> |Number of ISP Params|1|3|5|7|9|
> |-|-|-|-|-|-|
> |Proxy Approximation Accuracy of [27] (PSNR)|51.96|43.56|39.07|35.66|28.10|
> |Photo Finishing Tuning Quality of [27] (PSNR)|50.80|36.54|29.0|26.33|22.31|
>
> Since [27] recovers only 4 parameters while our experiment recovers 9, the finishing quality of is naturally lower in our paper. Additionally, ISP operations like sharpening are difficult for neural networks to approximate, as mentioned in [27], which further impacts performance.
>
> **Neural Proxy's Generalization**: The neural proxy learned in [27] struggles to generalize to unseen images, as shown in proxy accuracy drops on validation data:
>
> | |Train|Val|
> |-|-|-|
> |Proxy Approximation Accuracy (PSNR)|34.78|28.10|
>
> This explains the poorer performance in our additional HDR+ experiments.
>
> **Dataset used in [27] not released**. Therefore, the numbers reported by [27] are not reproducible. We trained this baseline on the FiveK dataset, following the details in [27].
> ### Monolithic Proxy [26]
> Our implementation reported a PSNR result above 20 on the FiveK dataset, comparable to the result reported by [27] when they reproduced this baseline. Like [27], [26] struggles with complex ISP operations and generalization. The vanishing gradient issue, analyzed in [27], further affects its performance.
> ### CMAES [18]
> **Reported PSNR Comparable to [27] Paper**: Our paper shows a PSNR above 28 for CMAES [18] on ISP tuning, comparable to the result reported by [27].
>
> **CMAES Performance with More Iterations**: In the table below, the performance of CMAES improves with more iterations, eventually reaching a PSNR of 32.12 on the FiveK dataset, comparable to RL's quality. While this indicates that our reproduction of the CMAES baseline is correct, the increased iterations lead to slower performance due to more queries to the ISP.
>
> |CMAES Iterations|10|100|200|500|
> |-|-|-|-|-|
> |PSNR|18.1|22.6|28.3|32.1|

---

> ### Author Response · Authors · 2024-08-12
>
> Dear Reviewer kK87,
>
> In our rebuttal, we address the implementation details of three baselines, explain the code used, and clarify terms in the paper, along with citations of CMAES (`Q1`). We also point out that certain behaviors of the baselines (such as the similar hue shift and green image in Figure 4) are due to the inherent flaws of the baselines (`Q2` `Q3`). Additionally, we provide more detailed experiments of the baselines to prove the correctness of our reproduction (`Q4`).
>
> We want to follow up to see if our responses address your concerns. We would be very grateful to hear additional feedback from you and will provide further clarification if needed.
>
> Thank you again for your time and effort.

---

> > ### Comment · Reviewer_kK87 · 2024-08-14
> >
> > The rebuttal and implementation details are sufficient.  I will raise my score to borderline accept.

---

### Official Review · Reviewer_syDC · 2024-07-13

**Soundness:** 3
**Presentation:** 3
**Contribution:** 3
**Rating:** 4
**Confidence:** 4

**Summary:**

Proposed Goal-Conditioned Reinforcement Learning for Photo Finishing Tuning. Specifically, the authors introduce a novel goal-conditioned reinforcement learning framework for parameter tuning in photo processing pipelines. Unlike existing methods, the proposed approach operates without relying on proxies and treats the pipeline as a black box. By leveraging a trained RL policy, it efficiently identifies optimal parameters in just 10 queries, contrasting with 500 queries typically required by zeroth-order methods. The proposed framework utilizes a goal image to guide iterative parameter tuning, allowing adaptation to diverse target images and styles. Experiments on photo finishing and stylization tasks validate the effectiveness and versatility of the proposed approach.

**Strengths:**

The paper is well written. It is easy to follow. The proposed method is well-motivated. The proposed method seems intuitive and effective for photo-finishing tuning tasks.

Empirical evaluations on image-based datasets show the efficacy of the proposed method.

**Weaknesses:**

The motivations behind some algorithmic choices are not clear.

Additional ablation studies would enhance the persuasiveness of the findings.

It would be interesting to see the performance of the proposed method on other datasets.

**Questions:**

(1) Why TD3? why not any other off-policy RL algorithm? Are there any specific reasons behind this choice?

(2) Enhancing the paper with more ablation studies would elevate its quality. For instance, exploring the impact of RL versus a greedy algorithm on performance would provide valuable insights.

**Limitations:**

Yes, the authors adequately addressed the limitations.

---

> ### Author Rebuttal · Authors · 2024-08-06
>
> # To reviewer syDC
> We sincerely thank you for reviewing our work. After reading your comments carefully, we summarize the following questions.
>
> ## Q1: Choice of RL Algorithm
> ***
> In our task, the choice of RL algorithm is not the primary factor driving performance; instead, our proposed state representation plays the most crucial role. We selected the TD3 algorithm due to its common usage and robustness to hyperparameters, but alternative algorithms like SAC or PPO would yield similar results.
>
> To support this, we performed an additional ablation study comparing different RL algorithms. We trained our photo-finishing policy using SAC and PPO algorithms, keeping the state representation, policy network, reward design, and termination conditions identical to our original TD3 implementation. All implementations were based on the stable-baselines3 library in PyTorch. We evaluated each algorithm's performance on the photo-finishing tuning task using the FiveK-Random dataset.
>
> **Table 1: RL Algorithm Comparison**
>
> |      | TD3   | SAC   | PPO   | TD3 w/o state representation |
> | ---- | ----- | ----- | ----- | ---------------------------- |
> | PSNR | 38.26 | 37.10 | 36.73 | 32.17                        |
>
> As shown above, different RL algorithms achieve comparable results on this task. However, removing the proposed state representation from the TD3 algorithm results in a significant performance drop, highlighting its importance. The results demonstrate that the state representation, not the specific choice of the RL algorithm, is the key factor behind our method's superior performance.
>
> ## Q2: Additional Ablation over Greedy Algorithm
> ***
> In our paper, we have shown that our RL-based method achieves higher photo-finishing quality and efficiency than optimization-based methods. We compared our method with baselines using zeroth-order search and first-order optimization with differential proxies but did not previously compare it with a simple greedy algorithm. Here, we provide an additional ablation study using a greedy algorithm.
>
> For the greedy algorithm, we followed Algorithm 1 from the [CRISP] paper, implementing a greedy search algorithm to find the optimal ISP parameters. This greedy algorithm mimics human behavior by incrementally adjusting each element towards the desired photo-finishing result until a stopping condition is met.
>
> > [CRISP] Learning Controllable ISP for Image Enhancement. TIP. 2023.
>
> As shown in Algorithm 1 in the **Pseudo Code for Greedy Algorithm** section below, the greedy algorithm mimics user behavior by incrementally improving image quality to achieve the desired results.  The algorithm starts with the ISP parameters at $s_{init}$ and iteratively adjusts them with a step size $t$ until the stopping condition $K$ is reached. We set each element in $s_{init}$ to be 0, step size $t=0.1$, and iterations $K=200$ to be consistent with other optimization-based methods. We evaluated the greedy algorithm on the FiveK-Target and HDR+ datasets, with the results shown in Table 1 of the rebuttal PDF. We also present the results in the table below.
>
> | Greedy Tuning Algorithm Evaluation | PSNR  | SSIM   | LPIPS  | Queries |
> | ---------------------------------- | ----- | ------ | ------ | ------- |
> | FiveK Target Photo Finishing Task  | 26.14 | 0.9250 | 0.1380 | 200     |
> | HDR+ Target Photo Finishing Task   | 25.79 | 0.9212 | 0.1542 | 200     |
>
> From the results, it can be observed that while the intuitive greedy algorithm achieves reasonable performance, it performs worse than the CMAES baseline under the same iterations. This indicates that the greedy strategy is not as effective as the evolutionary strategy in the CMAES algorithm. However, the greedy algorithm performs better than proxy-based methods [26, 27] because it requires no proxy training and searches directly in the parameter space. Note that like CMAES, the greedy tuning algorithm requires at least hundreds of queries to the ISP pipeline to converge, which are very time-consuming.
>
> Both greedy and CMAES are outperformed by our RL-based approach. This is because their searching processes are blind and brute-force, not conditioned on the input image, target image, or any prior on image processing operations. Our RL-based approach considers all these factors, requires no proxy training, and thus achieves better photo-finishing quality and efficiency.
>
>
>
> ## Q3: More Evaluation Dataset
> ***
> Please refer to Q1 in the **To All** section for the answer to this question. To summarize, we conducted further evaluations using the HDR+ dataset, demonstrating our RL-based framework's ability to generalize effectively to unseen datasets. Our method achieved a PSNR of 31.54 on the HDR+ photo-finishing tuning task, comparable to the 32.47 achieved on the FiveK dataset, and outperformed all baselines. This highlights our method's generalization capability across different datasets and its superior performance compared to other baselines, including CMAES, Greedy Search, Cascaded Proxy, and Monolithic Proxy.
>
>
>
> ## Pseudo Code for Greedy Algorithm
>
> ***
>
> $\textbf{Algorithm 1: Greedy Tuning Algorithm}$
>
> ***
>
> $\textbf{Inputs: } s_{\text{init}} \in \mathbb{R}^{D}, \text{ Step size } t, \text{ Stop condition } K, \text{ Input image } x, \text{ Tuning target image } \hat{x}$
> $\textbf{Initialization: } e \gets \infty, s \gets s_{\text{init}}, d \gets 1, i \gets 0, k \gets 0$
> $\textbf{While } k \leq K \textbf{ do}$
> $\quad \textbf{if } e < \text{MSE}(f_{\text{PIPE}}(x, s), \hat{x}) \textbf{ then}$
> $\quad \quad s_d \gets s_d - t, d \gets (d \bmod D) + 1, i \gets i + 1, k \gets k + 1$
> $\quad \quad \textbf{if } i = D \textbf{ then}$
> $\quad \quad \quad s \gets s + t, d \gets 1, i \gets 0$
> $\quad \quad \textbf{end if}$
> $\quad \textbf{else}$
> $\quad \quad e \gets \text{MSE}(f_{\text{PIPE}}(x, s), \hat{x}), i \gets 0, k \gets 0, s_d \gets s_d + t$
> $\quad \textbf{end if}$
> $\textbf{end while}$
>
> ***

---

> > ### Comment · Reviewer_syDC · 2024-08-13
> >
> > I appreciate the authors' response and the additional results provided in the rebuttal. However, I remain unconvinced by the results presented in Table 1, which actually raise more concerns.
> > 1. While I agree with the authors that state representation is crucial, the results showing variations with different RL algorithms suggest that the choice of RL algorithm is also important. It is unclear why TD3 performs better than the others. A stronger justification for the superiority of TD3 would be valuable.
> > 2. In RL experiments, it's essential to evaluate each method across multiple seeds since results can vary significantly. The table does not provide enough information to determine the best method conclusively. Reporting average results across multiple random seeds would be beneficial.
> > 3. The second table suggests that a naive greedy tuning algorithm performs well, though not as well as the RL-based method. It would be interesting to explore the possibility of developing a greedy tuning algorithm conditioned on the input image, target image, or prior image processing operations as a baseline.
> > 4. Although the authors have added more ablation studies in the rebuttal, they remain insufficient.
> > 5. While the authors have empirically demonstrated the method's generalization across different datasets, a detailed discussion on why the proposed method has superior generalization capability is necessary.
> > In summary, the problem is intriguing, and the proposed method appears effective, but it requires more in-depth analysis, additional ablation studies, and comparisons against stronger baselines to validate its effectiveness. Consequently, I have updated my score to borderline reject.

---

> ### Author Response · Authors · 2024-08-12
>
> Dear Reviewer syDC,
>
> In our rebuttal, we explain the reason for choosing TD3 and include further ablation studies to show that the choice of RL algorithm does not significantly impact performance. We also conduct more ablation over the greedy algorithm as suggested by the reviewer. Additionally, we test our method on the HDR+ dataset. Results demonstrate that our method generalizes well to unseen datasets.
>
> We want to follow up to see if our responses address your concerns. We would be very grateful to hear additional feedback from you and will provide further clarification if needed.
>
> Thank you again for your time and effort.

---

> ### Author Response · Authors · 2024-08-14
>
> Dear Reviewer syDC,
>
> Thank you for your reply. We would like to further discuss the questions with you below:
>
> ### Q1. why TD3 performs better
>
> In our task, the RL algorithm is only a tool for optimizing the policy network. We analyze why TD3 performs better as follows:
>
> TD3 is an off-policy RL algorithm with a deterministic policy. The deterministic policy allows it to explore the entire action space, including actions near the boundary, which is crucial for recovering boundary ISP parameters in our photo finishing task. TD3 also employs tricks like target policy smoothing and delayed policy updates to enhance robustness. In our task, actions like significantly brightening an image can cause large changes in the reward function, so these robustness tricks are crucial to the stability of the RL algorithm.
>
> In contrast, SAC uses a stochastic policy, sampling actions from a Gaussian distribution, which reduces the likelihood of exploring boundary actions, impacting performance. PPO also faces this issue with its stochastic policy. Both SAC and PPO also lack robustness tricks like target policy smoothing. Moreover, PPO is an on-policy algorithm, making it less sample-efficient in our complex photo finishing environment, potentially leading to suboptimal results.
>
> |      | deterministic policy | robustness tricks | off-policy |
> | ---- | -------------------- | ----------------- | ---------- |
> | TD3  | ✔                    | ✔                 | ✔          |
> | SAC  | ✗                    | ✗                 | ✔          |
> | PPO  | ✗                    | ✗                 | ✗          |
>
> These differences explain the variation in performance, but the effect of the RL algorithm is still minor compared to state representation. As shown in the first table of the rebuttal, TD3, SAC, and PPO all yield a PSNR of around 37 **with state representation**, whereas TD3 without state representation achieves only 32.17.
>
> ### Q2. multiple seeds
>
> We set the random seed to 1 in all our experiments in the paper and the rebuttal, ensuring fair comparison. In our main comparisons with all baselines, our method outperforms them by a large margin (over 4 dB in PSNR). Even in the ablation study over different RL algorithms, TD3 still outperforms SAC by 1.16 dB in PSNR. As discussed earlier, the RL algorithm is just a tool within our framework, and we are confident that we have selected the RL algorithm that best fits our task. Additionally, we plan to include the average scores over multiple seeds in the camera-ready version of this paper.
>
> ### Q3. greedy tuning performs well & stronger baseline with conditioning
>
> **There are no stronger baselines available.** We followed the setup from [27] for baselines in the photo finishing task. We compared our method to existing work in the field of photo finishing tuning, including CMAES [18] and proxy-based methods [26, 27]. Currently, there are no stronger baselines for the photo finishing tuning task.
>
> **Search-based methods (Greedy & CMAES) perform well because they optimize over more iterations.** Both the greedy tuning algorithm and CMAES are search-based methods. The search-based methods, including the greedy algorithm added in the rebuttal, only performs well when using 200 searching iterations, while our method can achieve better results with only 10 iterations. The main advantage of our RL method is its fast convergence speed, as shown in Table 2 in our paper.
>
> **Conditioning is not possible for greedy algorithm and is part of our contribution.** Traditional search-based methods, including greedy algorithms, are driven purely by the loss function's output relative to the parameters being adjusted. These methods lack the ability to condition on complex features like the characteristics of the input image, the target image, or prior processing operations. Such conditioning requires an understanding capability and the ability to generalize from past experiences, which is beyond the capabilities of traditional search-based algorithms.
>
> Developing a novel search algorithm conditioned on these complex characteristics is one of our contributions. Our proposed state representation encodes information about the input image, target image, and prior image processing operations. The policy network can then leverage this information to make decisions that are conditioned on these inputs, which leads to more efficient and effective optimization.
>
> ### Q4. why the proposed method generalizes well
>
> We have analyzed why our method generalizes better in Q1 (cross-dataset generalization) of the **To All** section in our rebuttal.

---

### Official Review · Reviewer_ASBs · 2024-07-15

**Soundness:** 2
**Presentation:** 3
**Contribution:** 3
**Rating:** 5
**Confidence:** 2

**Summary:**

This paper applies goal-conditioned reinforcement learning (RL) to photo finishing tuning. With only 10 queries, it demonstrates that goal-conditioned RL can achieve better performance than zeroth-order approaches that require 500 queries. Additionally, this method is non-differentiable, which might make it more suitable for commercial scenarios. Experiments show that this work can achieve better performance than previous efforts.

**Strengths:**

+ The first great application to adapt goal-conditioned RL to the photo finishing tuning task.
+ A novel design for the photo finishing state representation.
+ Achieved good performance on the evaluation benchmark.

**Weaknesses:**

- Some details are missing: The methods used in this work were trained on the training set, but it is unknown whether the compared methods also underwent the training stage.
- More evaluation on additional datasets is preferred.

**Questions:**

Please refer to the weakness section.

**Limitations:**

Please refer to the weakness section.

---

> ### Author Rebuttal · Authors · 2024-08-06
>
> # To Reviewer ASBs
>
> We sincerely thank you for reviewing our work. After reading your comments carefully, we summarize the following questions.
>
> ## Q1: Details on Baselines
>
> ***
>
> In our main paper, we compare three baselines: (1) CMAES: a zeroth order optimization method, that does not need training, (2) Monolithic Proxy: first-order optimization through a single monolithic neural network proxy, that requires training a proxy network, and (3) Cascade proxy: first-order optimization through multiple cascaded neural network proxies, also requires training proxy networks.
>
> Each of the baseline's implementation details is available in Q3 of **To All** section. Here, We briefly  summarize the principal of each of these baselines:
>
> 1. **CMAES** [18]: CMAES is a gradient-free search (zeroth order optimization) method using an evolution strategy. For our photo finishing tuning task, it directly optimizes the parameters of the image processing pipeline to match the target rendering style. Therefore, it does not undergo a training stage and optimizes directly on the validation set for multiple iterations instead. Such method faces no generalization challenges but converges very slowly as the searching processing is blind and brute-force. It also lacks exploration capability and is prone to stuck in local minimal with complex objectives.
> 2. **Monolithic Proxy** [26]: is a gradient-based optimization method (e.g. gradient descent). Since our task involves tuning an image processing pipeline (ISP) that is a non-differentiable black box, the gradient cannot flow through the ISP pipeline directly to drive the optimization of ISP parameters. Therefore, it requires training a neural network proxy to approximate the behavior of the ISP pipeline. The differential nature of the neural network proxy allows gradient flow to directly optimize ISP parameters during inference. However, for a complex image processing pipeline, this proxy is hard to train and may not fully reproduce original pipelines.
> 3. **Cascaded Proxy** [27]: is also a gradient-based optimization like [26], but the proxy neural network has a different architecture. The difference is that: the Monolithic Proxy uses a single UNet for the proxy, but the Cascaded Proxy improves the UNet with multiple small neural networks that are cascaded to address the vanishing gradient problem. Yet it still faces challenges with error accumulation, limited generalization capability of neural proxy, and cannot be applied to black-box pipeline.
>
> Please refer to Q3 in the **To All** section for further implementation details for these baselines.
>
> ## Q2: More Evaluation
>
> ***
>
> Please refer to Q1 in the **To All** section for the answer to this question. To summarize, we conducted further evaluations using the HDR+ dataset, demonstrating our RL-based framework's ability to generalize effectively to unseen datasets. Our method achieved a PSNR of 31.54 on the HDR+ photo-finishing tuning task, comparable to the 32.47 achieved on the FiveK dataset, and outperformed all baselines. This highlights our method's generalization capability across different datasets and its superior performance compared to other baselines, including CMAES, Greedy Search, Cascaded Proxy, and Monolithic Proxy.

---

> ### Author Response · Authors · 2024-08-12
>
> Dear Reviewer ASBs,
>
> In our rebuttal, we explain whether each baseline requires training. We also provide more evaluation on the HDR+ dataset. Results demonstrate that our method generalizes well to unseen datasets, outperforming all baselines by a large margin.
>
> We want to follow up to see if our responses address your concerns. We would be very grateful to hear additional feedback from you and will provide further clarification if needed.
>
> Thank you again for your time and effort.

---

### Author Rebuttal · Authors · 2024-08-07

# To ALL

We sincerely thank all reviewers for your comments. We summarize the following questions and add more results to the rebuttal PDF file.

## **Q1. Evaluation on more datasets (Reviewer ASBs syDC WirW)**
***

We test our RL-based framework directly on an extra dataset (HDR+ dataset). The results show that our method generalizes well to unseen datasets and outperforms all baselines. On the photo finishing tuning task, our RL-based framework achieves a PSNR of 31.54, comparable to 32.47 on the FiveK dataset, and outperforms first-order and zeroth-order optimization methods by a large margin.

**Dataset:** To further evaluate our method and demonstrate its generalizability, we used the HDR+ dataset:

> [HDR+] Burst photography for high dynamic range and low-light imaging on mobile cameras. TOG. 2016.

We used the official subset of the HDR+ dataset, which consists of 153 scenes, each containing up to 10 raw photos. The aligned and merged frames are used as the input, expertly tuned images serve as the photo-finishing targets.

**Evaluation result**: To test the generalization capability of our RL-based framework, we directly tested our pre-trained model on the new HDR+ dataset. We compare to baselines including CMAES, Cascaded Proxy, Monolithic Proxy, and Greedy Search (requested by reviewer syDC). In Table 1 of the rebuttal PDF file, we report PSNR, SSIM, LPIPS, and queries to the ISP pipeline. We have also included test results from the FiveK-Target dataset in the table's left column for easy comparison.

The results demonstrate that our RL policy generalizes effectively to unseen data, achieving higher photo-finishing quality than methods directly tuned on the test dataset. Qualitative comparisons in Figure 1 of the PDF file show our results are closer to targets, even with input and target images outside the training distribution.

**Cross dataset generalization**: In Table 1 of the PDF file, our RL-based method achieves a PSNR of 31.54 on the HDR+ photo-finishing task, comparable to 32.47 on MIT-Adobe FiveK. This shows that our RL policy, trained on the FiveK dataset, effectively generalizes to the HDR+ dataset. Such out-of-distribution capability is facilitated by our proposed photo-finishing state representation, which extracts invariant features for photo finishing, allowing adaptation to diverse inputs and goals beyond the training distributions.

The CMAES baseline shows consistent results on HDR+ compared to FiveK, as it is directly optimized on the test dataset without prior training. However, proxy-based methods [26, 27] perform worse because the proxy network trained on FiveK does not generalize well to HDR+, leading to incorrect gradients and poorer photo-finishing quality.

## **Q2. Clarification for terms cascaded and monolithic proxy (Reviewer kK87)**
***

Both [26] and [27] use neural network proxies to approximate ISP pipelines. The core difference is in the architecture of the neural network proxy: [26] uses a single UNet for the proxy, while [27] uses multiple small neural networks cascaded to address the vanishing gradient problem of UNet. The term "monolithic proxy" is borrowed from [27], which refers to [26] as a monolithic proxy. We refer to [27] as a "cascaded proxy" to differentiate their approaches.


## **Q3. More details on the implementation of baselines (Reviewer ASBs kK87)**
***

We provide details on the code used and implementation for each baseline. To summarize, we used as much of the provided code as possible and followed every publicly available detail.

### 3.1 Implementation details for cascaded proxy [27]

**Code used:** Although [27] provides code, it is incomplete and cannot be run directly. For instance, only the UNet architecture from [26] is specified in the code, not the cascaded network proposed in [27]. There are also known bugs in the GitHub repository (in GitHub issue #2) that prevent the pipeline from running. We used as much of the provided code as possible and reproduced other parts by following detailed descriptions in [27], as shown below.

**Proxy network training:** With no complete code available, we strictly followed the architecture in Tables 1 and 2 of [27] appendix, using 3 1x1 convolutions for pointwise ISP operations and 5 3x3 convolutions for areawise operations. We trained with the Adam optimizer, a learning rate of 1e-4, a batch size of 512, and 100 epochs, as recommended. Camera metadata was extracted from DNG files, as described in [27].

**Training dataset:** Since [27] does not release its dataset, we used the MIT-Adobe FiveK dataset, as in our method. Following Section 5 of [27], we used 1,000 raw images from FiveK and sampled 100 points for each ISP hyperparameter.

**ISP hyperparameter optimization:** After training the proxy neural network, we used it as a differential proxy to approximate the ISP pipeline. As in [27], we use the gradient flow through the proxy to directly optimize ISP hyperparameters, following [27]'s optimization details with the same Adam optimizer, same loss, same learning rate, and same iterations.

### 3.2 Implementation details for monolithic proxy [26]

**Code used:** [26] does not provide code, but [27] includes code for the UNet proxy used in [26]. We used this code for reproduction.

**Proxy network training:** The architecture uses a single UNet to approximate the ISP pipeline, with hyperparameters conditioned by concatenating extra planes to the features. We trained the proxy with the Adam optimizer, a learning rate of 1e-4, a batch size of 512, and 100 epochs.

The training set generation and first-order optimization method are the same as for [27].

### 3.3 Implementation details for CMAES [18]

CMAES [9,18] is a zeroth order optimization method without a neural proxy. As [18] does not provide code, we implemented CMAES using the `pymoo` library, enabling parallel execution on multi-core CPUs and achieving reasonable performance. This baseline does not require training.

---

### Decision · Program_Chairs · 2024-09-25

**Decision:**

Accept (poster)

**Comment:**

After a robust rebuttal period, the paper has been assigned the following ratings: weak accept, borderline accept (2x), and borderline reject. Two reviewers increased their ratings during the rebuttal process. The main lingering concerns from the borderline reject review relate to the motivation/justification for the selected RL algorithm and the value of including additional evaluation with more random seeds. The ACs agree that addressing these issues would strengthen the paper but do not believe they are sufficient to warrant rejection given the other strengths, including the magnitude of improvement over other approaches. The authors are encouraged to take the reviews and the post-rebuttal discussion into consideration as they prepare the camera-ready copy.